# ENHANCING ROBUSTNESS OF DEEP LEARNING VIA UNIFIED LATENT REPRESENTATION

## ABSTRACT

Adversarial examples and Out-of-Distribution (OoD) inputs constitute major problematic instances for the image classifiers based on Deep Neural Networks (DNNs). In particular, DNNs tend to be overconfident with their predictions, assigning a different category with a high probability. In this work, we suggest a combined solution to tackle both input types based on the Variational Autoencoder (VAE). First, we scrutinize the recent successful results in detecting OoDs utilizing Bayesian epistemic uncertainty estimation over weights of VAEs. Surprisingly, contrary to the previous claims in the literature, we discover that we can obtain comparable detection performance utilizing a standard procedure of importance sampling with the classical formulation of VAE. Second, we dissect the marginal likelihood approximation, analyzing the primary source of variation responsible for distinguishing inliers versus outliers, and establish a link with the recent promising results in detecting outliers using latent holes. Finally, we identify that adversarial examples and OoD inputs have similar latent representations. This insight allows us to develop separate methods to automatically distinguish between them by considering their non-similarities in the input space. The suggested approach enables pre-training a VAE model on specific input data, allowing it to act as a gatekeeper. This achieves two major goals: defending the DNN classifier against potential attacks and flagging OoDs. Once pre-trained, VAE can be plugged as a filter into any DNN image classifier of arbitrary architecture trained on the same data inputs without the need for its retraining or accessing the layers and weights of the DNN.

## 1 INTRODUCTION

Deep Neural Networks (DNNs) are applied to a rather diverse set of safety-critical tasks ranging from autonomous car driving to automatically-assisted medical diagnosis. However, the thorough theoretical foundation of deep learning is still lacking. It results in a limited understanding of how deep neural networks generalize. Such a situation led to the discovery of the following facts: $(i)$ there is a possibility to mislead the DNN classification with specifically forged inputs that, while preserving the semantics from the point of view of human observers, result in a wrong classification category by a DNN, i.e., the adversarial examples (Szegedy et al., 2013; Biggio et al., 2013; Goodfellow et al., 2014; Carlini & Wagner, 2016), and $(ii)$ the inability of the DNN to infer the fact that the provided input does not adhere to the data distribution they have been previously trained on, i.e., the overconfidence of DNN predictions with OoD inputs [1] (Nguyen et al., 2015; Hendrycks & Gimpel, 2016; Nalisnick et al., 2018).

The discriminative nature of the supervised image DNN classifiers implies learning a mapping from the input pixel space to the target labels. This mapping is usually considered to represent a categorical distribution over labels $y$ given the particular input $\mathbf{x}$: $p(y|\mathbf{x})$. However, in practice, the categorical distribution is based on the softmax activation function. As it has been recently formally proved, the softmax does not provide the desirable properties of categorical distribution and operates in a way similar to the k-means clustering, i.e., it partitions the transformed input space into several cones where every cone represents a different category (Hess et al., 2020). It may explain

---

[1]These inputs are commonly called outliers or Out-of-Distribution (OoD) inputs in the literature. Please note that we will use these terms interchangeably and consider them as synonyms.

why DNNs struggle with both adversarial examples and OoD: on one hand, it is feasible to find an adversarial direction from one category to another while attempting to preserve as little modification to the input as possible, especially for the examples that lie far from the cluster centroids, and on another hand when a new unseen data input arrives, the k-means clustering would necessarily cluster it into one of the categories resulting in overconfident predictions of OoD as in-distribution examples.

Recently, many different solutions have been proposed to address either the issue with adversarial examples (Zhang & Wang, 2019; Hu et al., 2019; Samangouei et al., 2018; Meng & Chen, 2017; Hwang et al., 2019) or the issue with OoD inputs (Daxberger & Hernández-Lobato, 2019; Ren et al., 2019; Hendrycks et al., 2018; Lee et al., 2017). The works, though, that solve both problems simultaneously within the same framework are few (Lee et al., 2018; Ahuja et al., 2019). These works are based on learning the DNN class-conditional weight uncertainties, which imply access to the model architecture, its weights, and output categories. Such an approach is closely interlinked with the DNN model under the protection, and it also introduces the unnecessary inductive bias by class conditioning. It makes the suggested methods non-modularizable and non-transferrable in a plug-and-play manner to other DNN architectures that require the same functionality of protection against adversarial attacks or OoD detection and that have been trained on the same input data.

Conversely, we apply an unsupervised Deep Generative Modeling (DGM) to tackle both of the problems, i.e., instead of learning discriminative mapping $p(y|\mathbf{x})$ and subsequently attempting to estimate the uncertainty of the weights under different inputs, DGM allows learning the approximation of a true distribution over the training data: $p(\mathbf{x})$ which in theory should assign a low density to the OoD and adversarial inputs. However, recent research revealed that such estimations are prone to errors, often providing higher likelihood values to both OoD and adversarial examples than to in-distribution data (Nalisnick et al., 2018).

To overcome this problem, we apply two recently suggested methods based on model parameter sensitivity analysis. (1) We use a *Bayesian* DGM, namely, VAE, that learns the weights uncertainty during training yielding the following posterior distribution: $p(\boldsymbol{\theta}|\mathcal{D})$ for the training data $\mathcal{D}$ over the model weights $\boldsymbol{\theta}$. It allows us to get an ensemble of the approximations of a true data distribution where each sample from the posterior $\boldsymbol{\theta} \sim p(\boldsymbol{\theta}|\mathcal{D})$ gives a separate instance of the model in the ensemble. Based on sampling from the posterior distribution, we estimate the likelihood of the input instance, however, instead of the usual calculation of the expected likelihood, we calculate the recently suggested scores of variance of the likelihoods between the different instance models in the ensemble (Glazunov & Zarras, 2022). The high degree of model/epistemic uncertainty is captured by the high values of the variance score. (2) We use recently suggested scores based on detecting if the corresponding latent code is in the hole or not (Glazunov & Zarras, 2023). We apply a single instance of classic VAE. Moreover, we enforce both compactness and continuity constraints on the latent representation and the corresponding encoder map. Overall, we suggest a single DGM based on VAE to detect both the OoD and adversarial inputs simultaneously, and we empirically evaluate the suggested approach based on the several datasets achieving promising results.

## 2 PROBLEM STATEMENT

### 2.1 ADVERSARIAL ATTACKS

There are two different perspectives on adversarial examples that give rise to two different definitions: one from the perspective of the generalization properties of the DNN and the other from the attacker's perspective. From the generalization perspective, an adversarial example (Szegedy et al., 2013) is a technique in which the input for the DNN image classier is intentionally modified to look almost the same as the original image to the human eye. Yet, it is perceived as something completely different by DNN. DNNs incorrectly classify such adversarial examples from the human perspective. On the other hand, the attacker perspective does not necessarily demand the part that relates to the imperceptibility of the difference (Biggio et al., 2013). On the contrary, if the miscreants want their attack's outcome to succeed, they should not constrain themselves to the superfluous imperceptibility demands. In this paper, we concentrate on the imperceptible examples. In addition, we analyze both alternatives from the perspective of their internal representation.

Furthermore, we conduct the experiments with both the adversarial examples generated for the discriminative model under attack and the adversarial examples generated to attack our defending VAE

filter. Specifically, for the former case, we consider three types of such attacks: Fast Gradient Sign Method (FGSM (FGSM) (Goodfellow et al., 2014), Carlini-Wagner (CW) attack (Carlini & Wagner, 2016), and Jacobian-based Saliency Map Attack (JSMA) (Papernot et al., 2016). For the latter case, we evaluate attacks on the encoder in the same vein as in Kuzina et al. (2024).

### 2.1.1 FAST GRADIENT SIGN METHOD

The FGSM attacks DNNs by leveraging their learning process based on gradients (Goodfellow et al., 2014). FGSM can be described by the following formula:

$$\mathbf{x}' = \mathbf{x} + \lambda \cdot \text{sgn}(\nabla_{\mathbf{x}}\ell(h_{\boldsymbol{\theta}}(\mathbf{x}), y_s)), \mathbf{x}' \in [0,1]^n$$

Here $\nabla_{\mathbf{x}}\ell$ is the gradient of the loss function w.r.t. the original input pixel vector $\mathbf{x}$, $y_s$ is the true or source label for $\mathbf{x}$, and $\boldsymbol{\theta}$ stands for the model parameters that are constant.

Gradient w.r.t. $\mathbf{x}$ is easier to calculate with backpropagation than for $\boldsymbol{\theta}$ which allows the fast generation of adversarial examples. FGSM exploits gradient ascent to increase the loss. Subsequently, the sign applies a max-norm constraint on the gradient value, and $\lambda$ represents a small magnitude of the step in the direction of increasing the loss. It represents the untargeted type of adversarial attacks.

FGSM can be converted into a targeted attack by substituting the source label with a target one $y_t$ and doing gradient descent instead of ascent, namely:

$$\mathbf{x}' = \mathbf{x} - \lambda \cdot \text{sgn}(\nabla_{\mathbf{x}}\ell(h_{\boldsymbol{\theta}}(\mathbf{x}), y_t)), \mathbf{x}' \in [0,1]^n$$

However, due to the fact that FGSM is designed to be fast rather than optimal, it is not necessarily guaranteed to produce the targeted adversarial examples of minimal perturbations.

### 2.1.2 CARLINI-WAGNER

Carlini-Wagner (CW) attack (Carlini & Wagner, 2016) aims at optimality in contrast with FGSM, i.e., it attempts to generate as little pixel noise as possible to succeed in the attack. It poses the following optimization objective:

$$\text{minimize } ||\varepsilon||_p \text{ subj. to } h_{\boldsymbol{\theta}}(\mathbf{x} + \varepsilon) = y_t, \ \mathbf{x} + \varepsilon \in [0,1]^n$$

where $\mathbf{x} \in [0,1]^n$ represents an image, $\varepsilon \in [0,1]^n$ is added noise to the image, and $y_t$ is a target class label of the image under attack. The noise level is calculated in terms of $L_p$ norms. Authors consider several norms; in this work, we concentrate on $L_2$-norm. This attack is one of the strongest known adversarial attacks.

### 2.1.3 JACOBIAN-BASED SALIENCY MAP ATTACK

JSMA (Papernot et al., 2016) leverages the saliency maps to devise an adversarial input. Namely, it computes the forward derivative of the whole DNN (Jacobian) w.r.t. the input, and based on this derivative, it constructs the saliency map. Large absolute values of the saliency map reveal the features that have a significant impact on the final output. The JSMA takes the maximum absolute value and perturbs it by a hyperparameter $\theta$ and repeats the process. The stopping criteria are either a successful attack with misclassification or reaching the total perturbation threshold of $\Upsilon$.

### 2.1.4 ATTACK ON ENCODER

This attack aims at maximization of the symmetric KL-divergence between the latent code of the reference input and the latent code of the reference input with the added perturbation:

$$\varepsilon = \arg \max_{||\varepsilon||_p \leq \delta} \text{SKL}\left[q(\mathbf{z}|\mathbf{x} + \varepsilon), q(\mathbf{z}|\mathbf{x})\right] \tag{1}$$

where $SKL$ is the symmetric KL-divergence, $\delta$ is the maximum amount of noise, and $q(\mathbf{z}|\mathbf{x})$ is the encoder under attack. The resulting adversarial perturbation is denoted as $\varepsilon$.

## 2.2 TRANSFERABILITY

It has been discovered that different architectures of DNNs trained to tackle the same classification problem on similar datasets tend to have similar fairly piece-wise linear decision boundaries that separate categories in the input data domain (Goodfellow et al., 2014). This property is called transferability. Transferability is especially dangerous since it allows to devise an adversarial attack that universally targets all DNNs with a similar final objective in a black-box manner (Papernot et al., 2016). Moreover, since we utilize a generative approach, we explore if there is transferability from the adversarial examples generated for a discriminative model to a generative one.

## 2.3 OUT-OF-DISTRIBUTION

Deploying a successful classifier requires from the system the ability to detect input data that are statistically anomalous or significantly different from those used in training. This is especially important for DNN classifiers since DNNs with the softmax classifier tend to produce overconfident predictions even for such Out-of-Distribution (OoD) inputs (Lee et al., 2018). The lack of reliability of DNN classifiers when faced with OoDs was recently addressed by various methods (Hendrycks & Gimpel, 2016; Hendrycks et al., 2018; Liang et al., 2017). According to recent research, the softmax activation function does not model a categorical distribution but represents a k-means clustering (Hess et al., 2020). That is why it seems logical to seek another approach. We decided to consider using unsupervised DGMs for that purpose. In our case, we apply the same VAE model to detect the OoDs based on the sensitivity analysis.

## 3 METHODOLOGY

We employ an approach based on the sensitivity analysis of the model parameter w.r.t. the different inputs. Namely, we test the level of stability of our model when dealing with OoDs versus IDs. There are two possible ways to achieve this goal. The first one is to utilize epistemic uncertainty estimation that would allow us to sample model parameters to be subsequently used for sensitivity analysis. The second one is to employ the learned posterior distribution over the latent codes in VAE and sample posterior for different latent codes. This approach does not change the parameters of DNNs used in the model, however, it allows conducting sensitivity analysis w.r.t. different sampled hypotheses from the latent posterior.

### 3.1 EPISTEMIC UNCERTAINTY IN OOD AND IN ADVERSARIAL ATTACKS

It has been shown that DGMs do not produce valid estimations of $p(\mathbf{x})$ when it comes to distinguishing between OoD and in-distribution (Nalisnick et al., 2018). Most of the results reveal DGMs being overconfident when dealing with OoD data. Another work dedicated to adversarial defense (Song et al., 2017) showed that it is possible to statistically differentiate between adversarial vs non-adversarial input data using DGMs. In this work, we first estimate the weight uncertainty to address this issue utilizing Bayesian and, in particular, variational inference.

### 3.2 ESTIMATION OF THE MARGINAL LIKELIHOOD

As suggested by Rezende et al. (2014), as soon as the VAE is trained, it is possible to estimate the likelihood of the input under the generative model using *importance sampling* w.r.t to the approximated posterior, namely:

$$p_{\boldsymbol{\theta}}(\mathbf{x}) \simeq \frac{1}{N} \sum_{i=1}^{N} \frac{p_{\boldsymbol{\theta}}(\mathbf{x}, \mathbf{z}_{(i)})}{q_{\boldsymbol{\phi}}(\mathbf{z}_{(i)}|\mathbf{x})}, \quad \text{where} \quad \mathbf{z}_{(i)} \sim q_{\boldsymbol{\phi}}(\mathbf{z}|\mathbf{x}) \tag{2}$$

However, as Nalisnick et al. (2018) discovered, we cannot rely directly on the likelihood estimations produced by a single DGM. This discovery is not surprising taking into consideration the fact that DGMs obtain the optimal parameters $\boldsymbol{\theta}^*$ under the Maximum Likelihood Estimation (MLE) for the $p(\mathcal{D}|\boldsymbol{\theta})$, where $\mathcal{D}$ represents the training data, resulting in a point estimate. Since in modern DNNs $|\boldsymbol{\theta}| \gg |\mathcal{D}|$, it is possible that there may be several models $\boldsymbol{\theta}$ that generated $\mathcal{D}$. Hence, it is impossible to estimate the epistemic uncertainty with a point estimate, which results in the inability of the model to provide a robust estimation of the likelihood for OoD and adversarial examples.

### 3.2.1 WEIGHT UNCERTAINTY: BAYES BY BACKPROPAGATION

Since we use variational inference to approximate our VAE posterior based on the assumption of the model with latent variables, we have chosen to apply the same variational approach to the weight uncertainty estimation instead of a point MLE estimate. Namely, we approximate the posterior distribution of the DGM parameters given the training data $p(\boldsymbol{\theta}|\mathcal{D})$ based on the method suggested by Blundell et al. (2015). This method initially was applied to the supervised learning, however, nothing prevents us from using it in the unsupervised setting. The ELBO objective is formulated in the following way:

$$\mathcal{L}_{\boldsymbol{\theta}}(\mathcal{D}, \lambda) = \int q(\boldsymbol{\theta}|\lambda) log(\frac{p(\boldsymbol{\theta})p(\mathcal{D}|\boldsymbol{\theta})}{q(\boldsymbol{\theta}|\lambda)})d\boldsymbol{\theta} \tag{3}$$

The approximation of the negative ELBO is obtained by:

$$-\widehat{\mathcal{L}_{\boldsymbol{\theta}}}(\mathcal{D}, \lambda) = \frac{1}{N}\sum_{i=1}^{N}\left[\log q(\boldsymbol{\theta}^{(i)}|\lambda) - \log p(\boldsymbol{\theta}^{(i)}) - \log p(\mathcal{D}|\boldsymbol{\theta}^{(i)})\right] \tag{4}$$

where $\boldsymbol{\theta}^{(i)}$ is sampled from the posterior $q(\boldsymbol{\theta}^{(i)}|\lambda)$.

We assume a diagonal Gaussian distribution for the variational posterior with parameters $\mu$ and $\sigma$. To make $\sigma$ to be always non-negative, we apply the same reparametrization as it was suggested by Blundell et al. (2015), namely $\sigma = \log(1 + \exp(\rho))$, yielding the following posterior parameters $\lambda = (\mu, \rho)$. For the prior, we also use the suggested scale mixture of two Gaussians:

$$p(\boldsymbol{\theta}) = \pi\mathcal{N}(\boldsymbol{\theta}|0, \sigma_1^2) + (1 - \pi)\mathcal{N}(\boldsymbol{\theta}|0, \sigma_2^2), \quad \text{where} \quad \pi = 0.5 \tag{5}$$

By adding weight uncertainty to the VAE, we are implementing *a Bayesian* VAE.

### 3.2.2 SCORES USED FOR PROBLEMATIC INPUTS DETECTION

After we approximated the variational posterior over the weights, the usual practice is to estimate the expected likelihood, the exact form of which can be formulated like this:

$$p(\mathbf{x}|\mathcal{D}) = \int p(\mathbf{x}|\boldsymbol{\theta})p(\boldsymbol{\theta}|\mathcal{D})d\boldsymbol{\theta} \tag{6}$$

The unbiased estimate of which can be obtained in the following way:

$$\mathbb{E}_{p(\boldsymbol{\theta}|\mathcal{D})}[p(\mathbf{x}|\boldsymbol{\theta})] \simeq \frac{1}{N}\sum_{i=1}^{N}p(\mathbf{x}|\boldsymbol{\theta}_i); \quad \text{where} \quad \boldsymbol{\theta} \sim p(\boldsymbol{\theta}|\mathcal{D}) \tag{7}$$

$p(\mathbf{x}|\boldsymbol{\theta}_i)$ is computed by importance sampling as in (2). As soon as the expected likelihood is estimated, one can apply some threshold that would distinguish if the considered input adheres to the in-distribution sample or not.

In this work, however, we aim to estimate the model parameter sensitivity. Hence, we calculate the sample standard deviation of the marginal log-likelihoods returned by the models within the ensemble:

$$\Sigma_{\boldsymbol{\Theta}}[\mathbf{x}] = \sqrt{\frac{1}{N-1}\sum_{\boldsymbol{\theta}\in\boldsymbol{\Theta}}(\log p(\mathbf{x}|\boldsymbol{\theta}) - \overline{\log p(\mathbf{x}|\boldsymbol{\theta})})^2} \tag{8}$$

It measures the variation within the log-likelihoods, so if there is a different level of sensitivity between the inliers and problematic inputs, then the standard deviation will capture this difference: the higher the value, the more uncertainty there is between the models about a particular input.

Furthermore, in the case of a single VAE, we instead apply the hole indicator. For this score we sample the approximated posterior $q_\phi(\mathbf{z}|\mathbf{x})$ with several latent codes $\mathbf{z}$ under a particular input $\mathbf{x}$ and compute the sample standard deviation of the log-likelihoods $\log p(\mathbf{x}|\mathbf{z})$:

$$\Sigma_{\mathbf{z}}[\mathbf{x}] = \sqrt{\frac{1}{N-1}\sum_{\mathbf{z}}\left(\log p(\mathbf{x}|\mathbf{z}) - \overline{\log p(\mathbf{x}|\mathbf{z})}\right)^2} \tag{9}$$

The higher the score, the farther the input is from the IDs.

Both of these scores allow for measuring the level of stability of the model w.r.t. different parameters. We can detect our problematic inputs based on the difference in this stability level.

### 3.2.3 Score for distinguishing between Adversarial and OoD Inputs

Note that utilizing the same score for both outliers and adversarial examples does not allow us to distinguish between them. To address this issue, we devise a simple algorithm for such a distinction.

---

**Algorithm 1** Active Defense Algorithm

---

**Require:** $\mathbf{x}$, $M$, $\boldsymbol{\theta}$, $\varnothing(\cdot)$, $h^{\text{ENC}}(\cdot)$, $h^{\text{DEC}}(\cdot)$, $\text{HMC}(\cdot)$, $\text{MSSSIM}(\cdot, \cdot)$
**Ensure:** Decision on whether $\mathbf{x}$ is an attack, an outlier, or an inlier
1: {Get a reconstruction and a latent code}
2: $\mathbf{z} \leftarrow h^{\text{ENC}}(\mathbf{x})$
3: $\mathbf{x}' \leftarrow h^{\text{DEC}}(\mathbf{z})$
4: {Check if $\mathbf{z}$ is in the hole}
5: **if** $\varnothing(\mathbf{z})$ **then**
6:     {Run active defense with $M$ steps}
7:     **for** $i = 1$ to $M$ **do**
8:         {One step of HMC}
9:         $\mathbf{z} \leftarrow \text{HMC}(\mathbf{z})$
10:     **end for**
11:     $\mathbf{x}_{\text{HMC}} \leftarrow h^{\text{DEC}}(\mathbf{z})$
12:     $\gamma_{\text{HMC}} \leftarrow \text{MSSSIM}(\mathbf{x}, \mathbf{x}_{\text{HMC}})$
13:     $\gamma_{\text{NO\_HMC}} \leftarrow \text{MSSSIM}(\mathbf{x}, \mathbf{x}')$
14:     {Check MSSSIM gain with a threshold $\boldsymbol{\theta}$}
15:     **if** $|\gamma_{\text{HMC}} - \gamma_{\text{NO\_HMC}}| > \boldsymbol{\theta}$ **then**
16:
17:         **return** "Attack"
18:     **else**
19:
20:         **return** "Outlier"
21:     **end if**
22: **else**
23:
24:     **return** "Inlier"
25: **end if**

---

Leveraging the intuition that adversarial examples also tend to land on the latent holes, it makes it possible to utilize the recently introduced approach for utilizing Hamiltonian Monte Carlo (HMC) to reevaluate the current latent code (Kuzina et al., 2024). If the generated image of the reevaluated latent code from the region close to the mean of the posterior is similar to the one that has been provided as the input, then it is highly likely to assume that there is an ongoing attack on the DNN. This similarity is based on Multi-Scale Structural Similarity (MSSSIM). This method represents an active defense approach.

The starting point is to identify if the corresponding latent code for the current input is located in the hole utilizing a hole indicator. If it is not in the hole, then we can immediately classify it as in-distribution input. Otherwise, the distinguishing between OoD and adversarial attack is implemented based on the restored latent code via HMC. The insight is that the resulting distance in the input space should be much closer for the adversarial inputs than for the OoDs (see Algorithm 1). As a result, we implement the robust VAE model against both outliers and adversarial examples with two levels of defense, allowing us to identify if we are being attacked or not.

### 3.2.4 Enforced Controlled Continuity and Compactness by Lipschitz continuity

To further increase robustness, we enforce a predefined Lipschitz constant on the encoder map of the VAE. First, it reduces the ability of the attacker to gain substantial benefits while generating adversarial examples with VAEs that possess encoding maps with great Lipschitz constants. Second, it allows the control of the properties of compactness of the mapped image to the latent space, which is beneficial for outlier detections utilizing latent holes. To that end, we employ the GroupSort activation function and enforce the corresponding Lipschitz constant (Anil et al., 2018).

### 3.3 Disentangling the variation and the Bayesian inference

We identify the source of the variation observed with Bayesian VAEs. The general procedure of the marginal likelihood estimation follows these steps:

1. Sampling the weights from the estimated posterior: $\boldsymbol{\theta} \sim p(\boldsymbol{\theta}|\mathcal{D})$.
2. Estimating the marginal likelihood for separate sampled models (Equation 2).
3. Computing a single value based on the separate estimated marginal likelihoods.

As it can be seen, there are two possible sources of variation, namely, variation from *Step 1* and variation from *Step 2*. It was hypothesized by Daxberger & Hernández-Lobato (2019) that the *Bayesian* inference over the DNN parameters is responsible for the observed variation of the results. In our work, we test if it is indeed the case by eliminating the first step and estimating the variation in the case of a simple classical VAE; namely, instead of sampling from $p(\boldsymbol{\theta}|\mathcal{D})$, we use a single VAE model that is used for marginal likelihood estimation several times. In such a case, all the variation comes only from the importance sampling. We apply the same scores for the Bayesian VAEs to identify if the variation persists for the classical VAEs.

### 3.3.1 DISSECTING THE SOURCE OF VARIATION

By taking the $\log$ of both sides of the Equation 2 and by factoring the joint probability $p_{\boldsymbol{\theta}}(\mathbf{x}, \mathbf{z})$, we can obtain the following equation for the importance sampling:

$$\log p_{\boldsymbol{\theta}}(\mathbf{x}) \simeq \frac{1}{N} \sum_{i=1}^{N} \left[ \log p_{\boldsymbol{\theta}}(\mathbf{x}|\mathbf{z}_{(i)}) + \log p(\mathbf{z}_{(i)}) - \log q_{\boldsymbol{\phi}}(\mathbf{z}_{(i)}|\mathbf{x}) \right] \tag{10}$$

All the scores that we have considered so far are measuring the variation of the left-hand side. To better understand where the variation comes from, we also consider the separate constituents of the right-hand side; namely, we measure standard deviations of all three terms separately, which allows us to identify the most uncertain term in the case of OoD detection.

### 3.4 ANALYZING LATENT REPRESENTATION

Our experiments confirm that adversarial examples can be identified using the same scores successfully applied to outliers. It implies that adversarial examples occupy latent holes similar to the OoDs. The difference is that it is possible to control the strength of the adversarial attack. Hence, we can visualize the dynamics of the attack strength w.r.t. the learned data representation in the latent space. To that end, we employ the learning procedure suggested by Jiang et al. (2017) to mold the latent data manifold into a mixture of Gaussians, the so-called Variational Deep Embeddings (VADEs). Such an approach allows us to calculate distances to the centroids of the learned clusters that can be visually inspected. In addition, no Lipschitz constraints are used for these experiments, so no restraints are applied for the adversarial locations.

## 4 EXPERIMENTS AND RESULTS

Our experiments have been conducted on several datasets widely used for validation of OoD and adversarial attacks, namely: MNIST(LeCun & Cortes, 2010), FashionMNIST(Xiao et al., 2017), SVHN(Netzer et al., 2011) and CIFAR10(Krizhevsky et al., 2010).

First, we estimated the impact of the number of dimensions of the latent space on the loss function. The dimensionality is closely connected with the dataset on which the model is trained. MNIST and FashionMNIST results reveal no particular need to exceed 10 latent dimensions since the loss function didn't significantly decrease after that value. For SVHN, we experimented with the number of latent dimensions up to 50, and the most optimal results were achieved with dimensionality equal to 20.

For our tests, we used two different architectures: for grayscale images, we applied a multilayer perceptron for both the encoder and decoder with two fully connected hidden layers. For RGBs images, we applied a Convolutional Neural Network (CNN) with two convolutional layers of 32 and 64 filters. For epistemic uncertainty estimation, all layers that contain parameters have been enhanced with the BBB, namely, convolutional 2D, fully connected, and convolutional 2D transpose. All the rest, such as reshape and flatten, are used with their default implementations as provided by the Tensorflow Keras (Chollet et al., 2015) framework. For a single VAE, we used the same architectures without the BBB. Moreover, the continuity of the encoder map is controlled via the specifically predefined Lipschitz constant calculated in the same way as in Glazunov & Zarras (2023) for the cases where the hole indicator is used.

All models have been trained for 1000 epochs. To evaluate the inputs, we sampled 100 different models for our ensemble. Since we have a doubly stochastic nature of the results, one due to the sampling from the latent posterior and the second one due to the sampling from the weights posterior, we ran the experiments 10 times each and averaged the final results.

For our implementation of BBB we noticed that random Normal initializer of the DNNs weights suggested as a prior in the original paper (Blundell et al., 2015) resulted in very slow convergence. So, to speed up the process, we also experimented with the following parameters: random Normal initializer with 0 mean and 0.1 standard deviation for $\mu$ and constant initializer for $\rho = -3$, which improved the training speed.

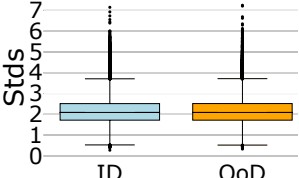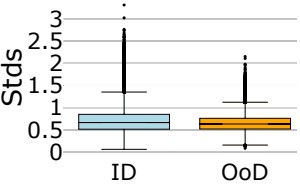

Figure 1: Standard deviations of the separate components of the ELBO *within* the importance samples for Fashion-MNIST as in-distribution (blue) vs MNIST as out-of-distribution (orange). **Left:** variation of the log-likelihood of the decoder $\log p(\mathbf{x}|\mathbf{z})$ **Middle:** variation of the encoder $\log q(\mathbf{z}|\mathbf{x})$. **Right:** variation of the latent prior $\log p(\mathbf{z})$.

The metrics that we used to validate both OoD are the area under Receiver operating characteristic (ROC) curve (ROC AUC), the area under the precision-recall curve (AUPRC), and the false-positive rate at 80% of true-positive rate (FPR80). We used two OoD benchmarks $(i)$ MNIST as in-distribution vs. FashionMNIST as OoD and $(ii)$ CIFAR10 as in-distribution vs. SVHN as OoD. As it can be observed from the results of Stds of LLs in Table 1, they are comparable with the state-of-the-art in the field (Daxberger & Hernández-Lobato, 2019).

Table 1: OoD detection results with Bayesian VAE based on Stds of LLs

| Metric | MNIST vs FashionMNIST | CIFAR10 vs SVHN |
|---|---|---|
| ROC AUC↑ | 99.76 | 90.88 |
| AUPRC↑ | 99.77 | 89.64 |
| FPR80↓ | 0.00 | 11.72 |

Subsequently, we performed experiments utilizing a single classical VAE testing if the previously observed variation persists. The obtained results demonstrate that variation that comes from the importance sampling is sufficient for the detection of the OoD inputs (see Table 2). It allows us to disentangle the variation from the Bayesian inference over the weights and directly use latent posterior sampling with a classical VAE.

Table 2: OoD detection results with classical VAE based on Stds of LLs

| Metric | MNIST vs FashionMNIST | CIFAR10 vs SVHN |
|---|---|---|
| ROC AUC↑ | 99.81 | 93.07 |
| AUPRC↑ | 99.82 | 91.23 |
| FPR80↓ | 0.00 | 11.36 |

In addition, we calculate the sample standard deviation of the separate terms on the right-hand side of Equation 10. The obtained values reveal the fact that most of the observed variance results from the likelihood term $\log p_{\boldsymbol{\theta}}(\mathbf{x}|\mathbf{z}_{(i)})$ that is parameterized by the decoder DNN. The boxplots of the standard deviations for all three terms (in the case of the classical VAE trained on the Fashion-MNIST dataset and tested on the MNIST as OoD) are plotted in Figure 1. As it can be seen, the variance obtained by the variational inference over the latent variable $q_{\phi}(\mathbf{z}|\mathbf{x})$ does not result in high values as one may have expected, which denotes that most of the responsibility for the variation is laid on the decoder which is more sensitive to the OoD inputs versus IDs. Such sensitivity has been observed for all of the considered datasets and models, which strongly supports the usage of the recently introduced hole indicator for the OoD input detection.

For the generation of the adversarial inputs, we used the Cleverhans framework (Papernot et al., 2018). We use the default discriminative DNN architecture for our victim classifier provided within this framework. We benchmark our model on three common attacks: FGSM, CW and JSMA (see Section 2.1 for more details). For FGSM, we used $\epsilon = 3$, for CW we used attack under $L_2$-norm, and we applied 128 attack iterations with 0.2 learning rate, and, finally, for JSMA we used $\theta = 1$ and $\gamma = 0.1$. For CW and JSMA we generated targeted attacks per each of 10 categories available in MNIST and FashionMNIST, for SVHN we applied an untargeted attack. In the case of FGSM, all inputs implemented an untargeted attack. Consequently, as Bayesian inference over DNN weights proves to be unnecessary, we employ a single VAE model, evaluating the results using a hole indicator (refer to Eq. 9).

Results of the experiments convincingly demonstrate that there is indeed transferability between discriminative and generative models. The adversarial examples generated for the classifier can be detected by the VAE, which is trained on the same dataset in an unsupervised manner (see Ta-

Table 3: Discriminative adversarial results: MNIST

| Metric | MNIST vs FGSM | MNIST vs CW | MNIST vs JSMA |
|---|---|---|---|
| ROC AUC↑ | 100.00 | 92.24 | 93.01 |
| AUPRC↑ | 100.00 | 92.55 | 90.13 |
| FPR80↓ | 0.00 | 11.72 | 10.16 |

Table 4: Discriminative adversarial results: FashionMNIST

| Metric | FMNIST vs FGSM | FMNIST vs CW | FMNIST vs JSMA |
|---|---|---|---|
| ROC AUC↑ | 96.49 | 95.01 | 83.97 |
| AUPRC↑ | 96.52 | 92.36 | 78.38 |
| FPR80↓ | 5.47 | 10.94 | 16.66 |

Table 7: MNIST: Multi-Scale Structural Similarity (MSSSIM)

| | No HMC | HMC | MSSSIM Gain |
|---|---|---|---|
| *Discriminative Adversarial Examples* | | | |
| MNIST FGSM $\varepsilon = 0.1$ | 0.43 | 0.34 | 0.09 |
| MNIST FGSM $\varepsilon = 0.3$ | 0.26 | 0.27 | 0.01 |
| MNIST CW | 0.18 | 0.20 | 0.02 |
| *Generative Adversarial Examples* | | | |
| MNIST $\varepsilon = 0.1$ | 0.43 | 0.85 | **0.42** |
| MNIST $\varepsilon = 0.2$ | 0.30 | 0.67 | **0.37** |
| MNIST $\varepsilon = 0.3$ | 0.25 | 0.64 | **0.39** |
| *Outliers* | | | |
| MNIST vs FMNIST | 0.03 | 0.09 | 0.06 |
| MNIST vs KMNIST | 0.21 | 0.16 | 0.05 |
| MNIST vs All White | 0.03 | 0.10 | 0.07 |

Table 8: FMNIST: Multi-Scale Structural Similarity (MSSSIM)

| | No HMC | HMC | MSSSIM Gain |
|---|---|---|---|
| *Discriminative Adversarial Examples* | | | |
| FMNIST FGSM $\varepsilon = 0.1$ | 0.28 | 0.17 | 0.11 |
| FMNIST FGSM $\varepsilon = 0.3$ | 0.19 | 0.24 | 0.05 |
| FMNIST CW | 0.33 | 0.26 | 0.07 |
| *Generative Adversarial Examples* | | | |
| FMNIST $\varepsilon = 0.1$ | 0.41 | 0.60 | **0.19** |
| FMNIST $\varepsilon = 0.2$ | 0.25 | 0.45 | **0.20** |
| FMNIST $\varepsilon = 0.3$ | 0.19 | 0.38 | **0.19** |
| *Outliers* | | | |
| FMNIST vs MNIST | 0.18 | 0.23 | 0.05 |
| FMNIST vs KMNIST | 0.20 | 0.19 | 0.01 |
| FMNIST vs All White | 0.21 | 0.17 | 0.04 |

bles 3 – 5). It is reproduced across a wide range of adversarial attacks and datasets. It is especially remarkable that they also tend to land to the holes in the VAE latent representation since they are detected based on the results of the hole indicator. Such a phenomenon may be explained by the similarity of internal representation within DNNs that are trained on the same datasets.

As can be observed from the results, the best values are achieved for the FGSM adversarial inputs, which result in a higher standard deviation of the log-likelihoods, leading to better detection. It seems not surprising, considering that FGSM does not aim at an optimal attack but the fastest one. CW, on the contrary, represents the least uncertainty, which also can be explained by the fact that this attack exploits the optimization procedure with the appropriate objective of as few modifications as possible to the input. JSMA is located somewhere in-between FGSM and CW.

Table 5: Discriminative adversarial results: SVHN

| Metric | SVHN vs FGSM | SVHN vs CW | SVHN vs JSMA |
|---|---|---|---|
| ROC AUC↑ | 86.74 | 77.35 | 82.75 |
| AUPRC↑ | 77.76 | 71.40 | 78.19 |
| FPR80↓ | 24.13 | 56.21 | 17.28 |

We evaluate the robustness of the proposed VAE filter by subjecting it to adversarial attacks designed explicitly for this model. We put under test a single VAE. The model is enforced with a controlled continuity on the encoder map considering ap-

Table 6: Generative adversarial examples

| Metric | Lipschitz MNIST: MNIST vs Adversarial | Lipschitz FMNIST: FMNIST vs Adversarial | Lipschitz MNIST heldout: MNIST 01 vs Adversarial |
|---|---|---|---|
| ROC AUC↑ | 97.89 | 93.40 | 99.98 |
| AUPRC↑ | 98.70 | 94.51 | 99.98 |
| FPR80↓ | 9.06 | 9.10 | 0.00 |

propriate properties of compactness of the latent image. As it can be seen from Table 6, the hole indicator successfully detects attacks on VAEs. It allows using only one score to detect both outliers and adversarial examples, including discriminative and generative ones.

Following this, we apply our algorithm based on active defense to distinguish between the outliers and both types of adversarial examples. Since the major value responsible for this distinguishing is based on MSSSIM gain, we register the corresponding values in Tables 7 and 8. It can be observed that generative adversarial examples can be easily discerned from the rest of the categories of problematic inputs. However, there is no possibility to delimit outlier and discriminative adversarial attacks relying only on the MSSSIM gain.

Finally, we visualize how the different attack strengths influence the location of adversarial latent codes within the learned data representation. This location is calculated w.r.t. the closest centroid of

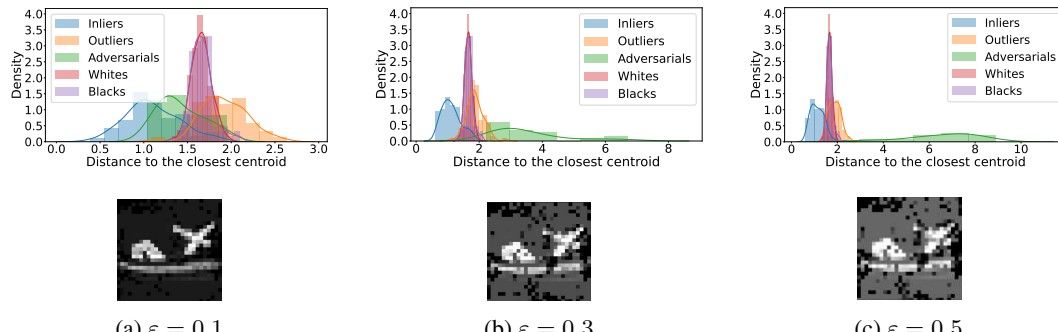

|                |                |                |
|:--------------:|:--------------:|:--------------:|
| (a) $\varepsilon = 0.1$ | (b) $\varepsilon = 0.3$ | (c) $\varepsilon = 0.5$ |

Figure 2: **From left to right**: The strength of FGSM attack, expressed by the magnitude of perturbations. **Top:** Distances to the closest centroid within the latent manifold for various categories of inputs. **Bottom:** Examples of a particular FashionMNIST instance that undergoes the corresponding strength of an attack.

the cluster to the corresponding adversarial latent code. As Figure 2 shows, the stronger the attack, the farther the corresponding latent codes drift away from the inlier manifold. Note that a weak adversarial attack is akin to the near-OoD instance, and a strong attack is akin to the far-OoD input.

## 5 DISCUSSION

The hole indicator confirms that transferability extends from discriminative to generative models, indicating a similar learned representation between those two approaches. Even though adversarial examples from the discriminative model end up in the latent holes of the VAE, the active defense through HMC cannot return to the regions with high probability. This suggests that despite some commonalities, differences still exist between discriminative and generative settings. Adversarial attacks on the VAE's latent space can be effectively distinguished from OoD inputs using active defense strategies. Furthermore, the internal latent representations of near- and far-OoD instances are similar to those of weak and strong adversarial attacks, respectively. Finally, contrary to common belief, Bayesian inference over DNN parameters is not essential for sensitivity analysis. We observe different levels of model stability w.r.t. inliers versus outliers, related to the differences in log-likelihood variances, revealing a connection with the recently introduced score of the hole indicator.

## 6 CONCLUSION

We explore two common types of problematic inputs in DNN classifiers: OoDs and adversarial attacks. Our proposed solution uses a variational autoencoder (VAE) to address both problems simultaneously. We initially evaluate the effectiveness of using Bayesian estimation of epistemic uncertainty from VAE weights to detect OoD inputs and discover that comparable results can be achieved by importance sampling with classical VAE formulations without resorting to Bayesian inference over weights. This result indicates that latent codes possess all the necessary information for measuring a model's sensitivity. Furthermore, we introduce a simple algorithm that distinguishes generative adversarial examples from both outliers and discriminative adversarial attacks using active defense. It enables identifying if the VAE model is currently being under attack. In addition, this algorithm allows for detecting both types of adversarial attacks: one is based on the imperceptible perturbations of the input image to the classifier, and it is based on the transferability of the adversarial examples from discriminative to generative models, while another is based on the attacks aimed at the encoder of the VAE. Finally, our approach allows a VAE model to be pretrained on specific datasets so that it functions as a filter, serving the purpose of protecting the DNN classifier from potential attacks and OoD inputs. This pre-trained VAE can be easily integrated as a filter with any DNN classifier, regardless of its architecture, trained on the same dataset, eliminating the need for further training or modification of DNN configurations.

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

# A APPENDIX

## A.1 CLASSICAL VAE'S OVERCONFIDENCE

As it was demonstrated by Nalisnick et al. in Nalisnick et al. (2018), all of the DGMs suffer from the overconfidence while trying to estimate the density of the out-of-distribution data assigning a higher density to the OoD inputs in comparison with ID data. We observed such an overconfidence during our experiments as well. A couple of examples of the overconfidence of the classical VAEs in our experimental setup can be seen in Figure 3.

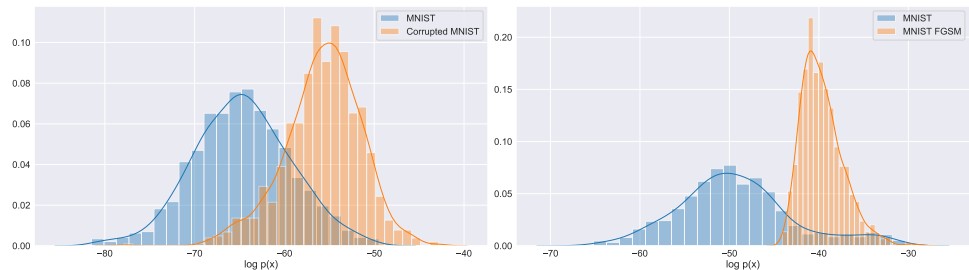

Figure 3: **Left:** Log-likelihoods for MNIST as in-distribution (blue) vs Corrupted MNIST as out-of-distribution (orange). **Right:** Log-likelihoods for MNIST as in-distribution (blue) vs MNIST FGSM attacks as out-of-distribution (orange).

## A.2 *Bayesian* VAEs VARIATION SCORING FOR THE REST OF OUR EXPERIMENTS

We ran out experiments also for MNIST as in-distrubtion vs Fashion-MNIST as OoD and for SVHN as in-distribution and CIFAR-10 as OoD. The results can be seen in Table 9 and Table 10.

Table 9: Scoring values across all types of *Bayesian* VAEs trained on MNIST data and tested on Fashion-MNIST as OoD

| | *MNIST vs. Fashion-MNIST* | | | | | | | | |
| | **BBB** | | | **SGHMC** | | | **SWAG** | | |
| | ROC AUC↑ | AUPRC↑ | FPR80↓ | ROC AUC↑ | AUPRC↑ | FPR80↓ | ROC AUC↑ | AUPRC↑ | FPR80↓ |
|---|---|---|---|---|---|---|---|---|---|
| **Expected LL** | 99.98 | 99.98 | 0.00 | 99.93 | 99.92 | 0.04 | **96.83** | **96.20** | **5.18** |
| **WAIC** | 99.99 | 99.99 | 0.00 | 99.94 | 99.94 | 0.02 | 80.37 | 76.25 | 33.56 |
| **Disagreement score** | 98.95 | 99.01 | 0.23 | 97.32 | 97.70 | 1.37 | 94.88 | 93.97 | 8.99 |
| **Entropy (ours)** | 99.42 | 99.47 | 0.02 | 98.50 | 98.75 | 0.29 | 95.72 | 95.20 | 8.37 |
| **Stds of LLs (ours)** | **99.99** | **99.99** | **0.00** | 99.91 | 99.91 | **0.00** | 80.37 | 82.78 | 39.12 |

Table 10: Scoring values across all types of *Bayesian* VAEs trained on SVHN data and tested on CIFAR-10 as OoD

| | *SVHN vs. CIFAR-10* | | | | | | | | |
| | **BBB** | | | **SGHMC** | | | **SWAG** | | |
| | ROC AUC↑ | AUPRC↑ | FPR80↓ | ROC AUC↑ | AUPRC↑ | FPR80↓ | ROC AUC↑ | AUPRC↑ | FPR80↓ |
|---|---|---|---|---|---|---|---|---|---|
| **Expected LL** | 58.65 | 61.79 | 77.72 | 57.09 | 60.56 | 80.18 | 58.98 | 62.06 | 76.52 |
| **WAIC** | 64.46 | 66.01 | 68.39 | 62.17 | 64.38 | 72.45 | 62.84 | 68.42 | 75.25 |
| **Disagreement score** | 85.20 | 88.35 | 30.26 | 85.31 | 88.52 | 28.66 | 77.58 | 80.36 | 45.60 |
| **Entropy (ours)** | 87.80 | 90.63 | 20.77 | 87.89 | 90.76 | 19.91 | **80.01** | **83.24** | **41.58** |
| **Stds of LLs (ours)** | **93.29** | **91.51** | **10.99** | **94.70** | **93.95** | **8.67** | 59.31 | 53.36 | 61.78 |

## A.3 HAMILTONIAN MONTE CARLO ALGORITHM

We employ the same approach as suggested in Kuzina et al. (2024).

In the Hamiltonian Monte Carlo (HMC) framework, the target distribution is given by the product of $p(\mathbf{x}|\mathbf{z})$ and $p(\mathbf{z})$. The Hamiltonian represents the energy of the combined distribution of $\mathbf{z}$ and the auxiliary variable $\mathbf{p}$, defined as follows:

$$H(\mathbf{z}, \mathbf{p}) = U(\mathbf{z}) + K(\mathbf{p}),$$

where

$$U(\mathbf{z}) = -\log p_{\boldsymbol{\theta}}(\mathbf{x}|\mathbf{z}) - \log p(\mathbf{z}),$$

and

$$K(\mathbf{p}) = -\frac{1}{2}\mathbf{p}^T\mathbf{p}.$$

For the corresponding pseudocode for restoring the latent code, please see following Aglorithm 1.

---

**Algorithm 1:** A single iteration of HMC

---

**Input: z**, $\eta, L$

// Sample the auxiliary variable
$\mathbf{p} \sim \mathcal{N}(\mathbf{0}, \mathbb{I})$
$\mathbf{z}^{(0)} := \mathbf{z}, \mathbf{p}^{(0)} := \mathbf{p}$

// Make $L$ steps of leapfrog
**for** $l = 1$ **to** $L$ **do**
$\quad \mathbf{p}^{(l)} := \mathbf{p}^{(l-1)} - \frac{\eta}{2}\nabla_{\mathbf{z}}U(\mathbf{z}^{(l-1)})$
$\quad \mathbf{z}^{(l)} := \mathbf{z}^{(l-1)} + \eta\nabla_{\mathbf{p}}K(\mathbf{p}^{(l)})$
$\quad \mathbf{p}^{(l)} := \mathbf{p}^{(l)} - \frac{\eta}{2}\nabla_{\mathbf{z}}U(\mathbf{z}^{(l)})$

// Accept new point with probability $\alpha$
$\alpha := \min\left(1, \exp\left(-H(\mathbf{z}^{(L)}, \mathbf{p}^{(L)}) + H(\mathbf{z}^{(0)}, \mathbf{p}^{(0)})\right)\right)$

$\mathbf{z} := \begin{cases} \mathbf{z}^{(L)} & \text{with probability } \alpha, \\ \mathbf{z}^{(0)} & \text{otherwise.} \end{cases}$

**return z**

---

### A.4 CLASSICAL VAEs VARIATION SCORING FOR THE REST OF OUR EXPERIMENTS

Table 11: Scoring values for the classical VAEs trained on MNIST and Fashion-MNIST data

| | *Classical VAE* | | | | | |
| | **MNIST vs. Fashion-MNIST** | | | **Fashion-MNIST vs. MNIST** | | |
| | **ROC AUC↑** | **AUPRC↑** | **FPR80↓** | **ROC AUC↑** | **AUPRC↑** | **FPR80↓** |
|---|---|---|---|---|---|---|
| **Expected LL** | **99.97** | **99.97** | **0.00** | 46.72 | 51.54 | 92.57 |
| **WAIC** | 99.96 | 99.96 | 0.00 | 64.07 | 64.43 | 66.98 |
| **Disagreement score** | 97.86 | 98.09 | 1.11 | 96.83 | 97.56 | 0.84 |
| **Entropy (ours)** | 98.67 | 98.84 | 0.38 | 98.18 | 98.63 | **0.08** |
| **Stds of LLs (ours)** | 99.81 | 99.82 | 0.00 | **99.68** | **99.64** | 0.36 |

