# OpenReview forum: "Enhancing Robustness of Deep Learning via Unified Latent Representation"
_ICLR.cc/2025/Conference — ICLR 2025 Conference Withdrawn Submission_

### Official Review · Reviewer_X3e2 · 2024-10-28

**Soundness:** 1
**Presentation:** 1
**Contribution:** 1
**Rating:** 1
**Confidence:** 4

**Summary:**

Summary:
This paper presents a unified framework utilizing Variational Autoencoders (VAEs) to concurrently detect adversarial examples and Out-of-Distribution (OoD) inputs, without necessitating modifications to or access to the protected classifiers trained on the same dataset. Specifically, through scrutinizing the score based on Bayesian epistemic uncertainty for OoDs detection, the authors demonstrate that comparable detection performance can be achieved using an alternative score based on importance sampling with classical VAE formulations, thus circumventing the need for Bayesian approaches. Furthermore, they confirm that adversarial examples can be identified using the latter scores. After that, they introduce techniques to distinguish adversarial from OoD inputs by analyzing latent space and applying the Multi-Scale Structural Similarity (MSSSIM). Finally, the experiments conducted on several datasets widely used for validation of OoD and adversarial attacks demonstrate the effectiveness of their approach. However, the motivation of the overall method design is not clear, and the writing of the paper needs to be significantly improved.

**Strengths:**

1.	The authors propose a VAE-based framework, in which the VAE is trained in an unsupervised manner on the training data of protected classifiers, to simultaneously detect adversarial examples and Out-of-Distribution (OoD) inputs.
2.	They demonstrate that an alternative score based on importance sampling with classical VAE can achieve comparable detection performance for OoD inputs compared to Bayesian methods.
3.	They conduct extensive experiments on several widely used datasets for validation of OoD and adversarial attacks, demonstrating that the proposed method can effectively detect adversarial examples and OoD inputs.

**Weaknesses:**

1.	In the introduction, the authors present their contributions with a focus that is not entirely consistent with the emphasis in the abstract. In the abstract, they highlight their discovery that Bayesian VAE methods underperform compared to classic VAE methods. However, in the introduction, they describe their contributions as the application of both Bayesian VAE and classic VAE methods for detecting adversarial and out-of-distribution (OoD) samples. Additionally, they do not mention in the introduction a separate method they propose for automatically differentiating between adversarial examples and OoD inputs.
2.	The abstract does not effectively convey the authors' contributions to the reader. In the abstract, the authors claim to have developed a distinct method to automatically differentiate between adversarial examples and out-of-distribution (OoD) inputs. However, as demonstrated in Tables 7 and 8 in the experimental section, the method proposed in Section 3.2.3 is only capable of distinguishing generative adversarial examples against VAEs from other problematic inputs and does not effectively separate outliers from discriminative adversarial attacks against classifiers.
3.	Although the authors extend the out-of-distribution (OoD) detection score from [1] to simultaneously detect both adversarial examples and OoD inputs and design a method based on [2] to differentiate generative adversarial examples for VAEs from OoD samples, they do not provide sufficient theoretical insights into the validity of these proposed methods.
4.	In the methods section, the paper devotes considerable space to explaining the rationale behind the Bayesian VAE-based approach and its associated scoring method. However, the authors ultimately favor the classic VAE-based approach, yet provide only a brief and limited introduction to it. This imbalance makes it difficult for readers to achieve a consistent and coherent reading experience.
5.	The methods section of the paper is divided into numerous subsections, some of which lack clarity in conveying their intended content. Additionally, the logical connections between these subsections are not well-defined, making it challenging for readers to grasp the core content of the paper.
6.	Many false causal claims in the paper, eg., " However, the thorough theoretical foundation of deep learning is still lacking. It results in a limited understanding of how deep neural networks generalize. Such a situation led to the discovery ……". The authors should carefully check these.
7.	Many typical adversarial and OOD detection methods, such as [1-10], are missing. It would be better for the authors to incorporate these works in Related work.

[1] Characterizing Adversarial Subspaces Using Local Intrinsic Dimensionality, ICLR 2018.

[2]  Adversarial Example Detection Using Latent Neighborhood Graph, ICCV 2021.

[3] LiBRe: A Practical Bayesian Approach to Adversarial Detection, CVPR 2021.

[4] Detecting Adversarial Data by Probing Multiple Perturbations Using Expected Perturbation Score, ICML 2023.

[5] IntensPure: Attack Intensity-aware Secondary Domain Adaptive Diffusion for Adversarial Purification, IJCAI 2024.

[6] A Baseline for Detecting Misclassified and Out-of-Distribution Examples in Neural Networks, ICLR 2017.

[7] Out-of-distribution detection using multiple semantic label representations. NIPS 2018.

[8] A simple unified framework for detecting out-of-distribution samples and adversarial attacks, NIPS 2018.

[9] On the Importance of Gradients for Detecting Distributional Shifs in the Wild, NIPS 2021.

[10] Unsupervised Out-of-Distribution Detection with Diffusion Inpainting, ICML 2023.

8.	In the reference list, some entries are missing the conference or journal in which the papers were published, while others correspond to works that have since been formally published. Therefore, the reference list requires a thorough review. Below are some problematic references:

a. Sorting Out Lipschitz Function Approximation. ICML 2019.

b. Towards Evaluating the Robustness of Neural Networks. IEEE Symposium on Security and Privacy 2017.

c. Explaining and Harnessing Adversarial Examples. ICLR 2015.

d. A Baseline for Detecting Misclassified and Out-of-Distribution Examples in Neural Networks. ICLR 2017.

e. Deep Anomaly Detection with Outlier Exposure. ICLR 2019.

f. Alleviating Adversarial Attacks on Variational Autoencoders with MCMC. NeurIPS 2022.

g. PuVAE: A Variational Autoencoder to Purify Adversarial Examples. IEEE Access 2019.

h. Training Confidence-calibrated Classifiers for Detecting Out-of-Distribution Samples. ICLR 2018.

i. Enhancing The Reliability of Out-of-distribution Image Detection in Neural Networks. ICLR 2018.

j. MagNet: A Two-Pronged Defense against Adversarial Examples. CCS 2017.

k. Do Deep Generative Models Know What They Don't Know? ICLR 2019.

l. Practical Black-Box Attacks against Machine Learning. AsiaCCS 2017.

m. Defense-GAN: Protecting Classifiers Against Adversarial Attacks Using Generative Models. ICLR 2018.

n. PixelDefend: Leveraging Generative Models to Understand and Defend against Adversarial Examples. ICLR 2018.

**Questions:**

1.	In the abstract, the authors claim that a pre-trained VAE can be seamlessly integrated into any architecture of a deep neural network (DNN) image classifier for the detection of adversarial examples and out-of-distribution (OoD) inputs. However, the experimental section lacks a description of the architecture of the image classifier used. Therefore, could the authors provide additional details about the architecture of the protected image classifier? Furthermore, could the authors validate the performance of the same pre-trained VAE for detection across a broader range of DNN image classifiers with diverse architectures?
2.	Why do the experimental results presented in Table 9 of the appendix conflict with those in Table 1? Additionally, why does Table 1 not include the performance of baseline methods, as seen in Table 9? In Table 9, there is a comparative method labeled 'Entropy (ours)', yet this method is never mentioned elsewhere in the paper. Furthermore, while the authors state in the abstract that their results using the Bayesian VAE method are inconsistent with those of prior work, why do the results in Tables 9 and 10 align with the results presented in Tables 3 and 4 of [3]?
3.	For the Bayesian VAE-based out-of-distribution (OoD) detection method, could the authors provide experimental results for the scenario of using FashionMNIST as in-distribution data versus MNIST as OoD data? Furthermore, for the classical VAE-based OoD detection method, could the authors provide experimental results for the scenario of using SVHN as in-distribution data versus CIFAR10 as OoD data?
4.	Could the authors provide a detailed explanation of the specific methods used to generate generative adversarial examples?
5.	The paper states that the experimental performance of the Bayesian VAE method is inferior to that presented in [3]. Could the authors provide the specific codebase used for their Bayesian VAE implementation? Additionally, could they check for any differences compared to the implementation in [3] and offer some explanations for these discrepancies?
6.	Given that each experiment in the experimental section is conducted 10 times to account for randomness and then the authors use the average as a final result, could the authors provide the corresponding variance for the relevant experimental outcomes?
7.	What is your motivation for your "first method" mentioned in the Abstract? And what is your view to be validated?
8.	What are "latent holes" and why does the "The suggested approach" allow it to be a "gatekeeper" in the Abstract?
9.	Could the authors visualize the dynamics of the attack strength with respect to the learned data latent representation using Variational Deep Embeddings (VADEs), as mentioned in Section 3.4 of the paper? Additionally, could they provide some insights and analysis related to this visualization?

[1] Vacant Holes for Unsupervised Detection of the Outliers in Compact Latent Representation. UAI 2023.

[2] Alleviating Adversarial Attacks on Variational Autoencoders with MCMC. NeurIPS 2022.

[3] Do Bayesian Variational Autoencoders Know What They Don’t Know?UAI 2022.

**Details Of Ethics Concerns:**

Please see my detailed comments above.

---

### Official Review · Reviewer_CfSu · 2024-11-02

**Soundness:** 3
**Presentation:** 2
**Contribution:** 2
**Rating:** 5
**Confidence:** 4

**Summary:**

This paper proposes a unified framework based on Variational Autoencoders (VAE) to enhance deep learning robustness by simultaneously detecting adversarial and out-of-distribution (OOD) samples. The method provides an innovative solution for modular plug-and-play integration without altering the original model architecture, aiming to improve resilience across various applications.

**Strengths:**

The proposed VAE-based unified framework effectively combines adversarial and OOD sample detection, providing innovation and practicality. This approach offers a comprehensive protection solution for the relevant field. By acting as a modular plugin without modifying the original model’s architecture or weights, the method enhances adaptability and compatibility.

**Weaknesses:**

1. From a readability perspective, the paper lacks detailed derivation processes. I could not find corresponding derivations in the appendix, and extensive literature review is required to understand the method’s contents. This is unreasonable and makes the paper difficult to follow.

2. I have concerns about the method’s practicality, as detecting a single sample requires a computation cost that is significantly higher than the original task, which is generally unacceptable in real-world applications. The paper needs to provide more analysis on the performance and overhead of the primary task to demonstrate the feasibility of the algorithm.

3. Additionally, I am concerned about the lack of detailed VAE training specifics (at least to a reproducible extent), which is crucial for both the algorithm’s foundation and effectiveness.

4. The dataset used in the experiments is relatively simple. If using ImageNet for computation and validation is challenging, at least including CIFAR-100 results as a reference would be beneficial.

5. It is necessary to include experiments comparing with OOD detection and adversarial attack detection algorithms (demonstrating the main point that these methods cannot detect both simultaneously).

**Questions:**

See Weaknesses

---

### Official Review · Reviewer_Bpz5 · 2024-11-02

**Soundness:** 2
**Presentation:** 2
**Contribution:** 1
**Rating:** 1
**Confidence:** 4

**Summary:**

The paper addresses two challenges including adversarial examples and OOD inputs. The authors propose a solution using VAEs to those issues. The authors use some form of scores detecting if the corresponding latent code is in the hole or not. Their found that adversarial and OOD inputs share similar latent representations in VAEs. Their pre-trained VAE can be potentially used as a filter with any DNN classifier architecture trained on the same data.

**Strengths:**

Their solution based on VAEs can potentially detect adversarial inputs and OOD inputs. Their insights adversarial inputs and OOD inputs have some similarities in the latent space are interesting, which can be potentially useful for other researchers.

**Weaknesses:**

* The authors state "...recent research revealed that such estimations are prone to errors, often providing higher likelihood values to both OoD and adversarial examples than to in-distribution data (Nalisnick et al., 2018)." and "It has been shown that DGMs do not produce valid estimations of p(x) when it comes to distinguishing between OoD and in-distribution (Nalisnick et al., 2018)." However, the paper does not summarize the recent research results in the area. Indeed, there have been numerous advanced schemes such as De-biasing (DB), Input Complexity Based Likelihood, Likelihood Ratio (LRat), Likelihood Regret (LReg), Watanabe Akaike Information Criterion (WAIC), which are most based on VAEs.

* I do not understand why "Bayesian" generative models must be used. There are lots of other types of generative models that are working very well such as GANs, flow-based models, autoregressive models, diffusion models, energy based models, etc.

* I do not understand why VAE must be particularly considered. As discussed above, as of today, VAEs are fairly weak generative models.

* Developing their specific mechanisms, the authors particularly chose the results form some particular researchers, Glazunov & Zarras, 2022 & 2023. But, I am not sure why those particular results are leveraged here.

* My major concern is about the datasets. Only very small datasets are tested such as MNIST, FashionMNIST, SVHN, and CIFAR-10. These days, in numerous works on OOD, larger datasets are considered such as CIFAR-100, tiny-ImageNet, mini-ImageNet. Furthermore, most recently, in many works, ImangeNet-1k is often considered as in-distribution, while iNaturalist and Texture are considered as OOD.

* The baseline methods are also very concerning. I hardly see any performance comparison with any other schemes. Seems that the authors tested their own schemes without any comparison to others in the literature. In addition to the methods that I mentioned above, over the past a couple of years, a number of very effective OOD methods have been developed. For example, please see the following references:

Yang et al., "Generalized Out-of-Distribution Detection: A Survey," 23 Jan 2024.

Miyai et al., "Generalized Out-of-Distribution Detection and Beyond in Vision Language Model Era: A Survey," 31 Jul 2024.

**Questions:**

* Strong justifications are necessary why VAEs are used instead of numerous highly performing generative models.

* Their particular method should be better justified.

* Much larger datasets should be used such as ImangeNet-1k.

* Extensive performance comparisons should be performed with other methods.

---

### Official Review · Reviewer_w1i3 · 2024-11-04

**Soundness:** 1
**Presentation:** 2
**Contribution:** 1
**Rating:** 1
**Confidence:** 4

**Summary:**

The paper introduces a VAE-based method to tackle both OOD detection and adversarial robustness. They use a hole indicator obtained from sampling from $q(z|x)$ as the score for distinguishing clean from out-of-distribution and adversarial inputs. The paper further uses distance in the input space to distinguish adversarial from OOD inputs. The encoder part of the VAE is further trained with GroupSort activation and pre-defined Lipschitz constant as is commonly done in the field of provable robustness. Evaluation is done on Mnist, FashionMnist, SVHN and CIFAR10.

**Strengths:**

- The Probabilistic modeling seems reasonable

**Weaknesses:**

- Dataset selection is not acceptable for an ICLR2025 paper. MNIST, FashionMNIST, and SVHN are only toy datasets. None of the datasets has a sufficient number of classes. CIFAR100 would be the bare minimum to see how well the method scales with the total number of classes. ImageNet-1K would be ideal since there is no high-resolution dataset (above 32x32) in the current pool.
-  The OOD evaluation is trivial. As many works have shown (e.g. [Breaking Down Out-of-Distribution Detection](https://arxiv.org/abs/2206.09880)), the only challenging benchmarks are  OOD benchmarks that require the model to distinguish classes (for example CIFAR10 vs CIFAR100) instead of dataset statistics. CIFAR10 vs LSUN, CIFAR100, and CelebA is the minimum requirement (and similarly for CIFAR100).
Other works like [PViT: Prior-augmented Vision Transformer for Out-of-distribution Detection](https://arxiv.org/html/2410.20631v1) are doing much more challenging benchmarks like ImageNet vs iNat, NINCO and ImageNet-o.
- "Bayesian variational autoencoders for unsupervised out-of-distribution detection" is not even close to "state-of-the-art" in the field. Density-based methods have been shown time and time again to be worse at OOD detection than classifier-based methods. State-of-the-art OOD has been held by some form of pre-trained ViT for years, for example, [PViT](https://arxiv.org/html/2410.20631v1) or [Exploring the Limits of Out-of-Distribution Detection
](https://arxiv.org/pdf/2106.03004). Just because this method is based on VAEs does not mean that it does not have to compare to actual state-of-the-art methods based on different methods.
- Cleverhans is severely outdated for the evaluation of adversarial robustness. [AutoAttack](https://github.com/fra31/auto-attack) / [RobustBench](https://robustbench.github.io/) is the standard used by almost every paper in the field.
- Adversarial robustness results are given in isolation. Please compare them to state-of-the-art values in this field from the RobustBench leaderboard. Also, from Section 3.2.3 it seems like the paper can only detect adversarial examples but not actually classify them.
- The paper does not consider adversarial attacks on OOD data (as was done in [this paper](https://arxiv.org/abs/2003.09461)). This is commonly done to make an OOD sample appear to be from the in-distribution using adversarial attacks. Instead, the paper seems to distinguish only between clean data, OOD data, and adversarial data.
- There seem to be few methodological contributions in this paper and instead, the paper mostly combines existing ideas in a rather straightforward fashion.

**Questions:**

- Did you try scaling this method? What are the challenges with applying this to much larger datasets?
- What is the impact of Lipschitz regularisation and GroupSort activations?

---

### Official Review · Reviewer_YAdT · 2024-11-06

**Soundness:** 4
**Presentation:** 3
**Contribution:** 4
**Rating:** 8
**Confidence:** 3

**Summary:**

The authors address the problem of distinguishing between adversarial inputs and out-of-distribution (OOD) inputs to a VAE.  They propose a methodology for identifying whether an input is adversarial or OOD that does not rely on Bayesian inference over the model weights.  Bayesian inference is taken as the state of the art. They compare their methodology to the state of the art, and demonstrate its effectiveness using pre-defined in-distribution and out-of-distribution datasets, and again using three different kinds of adversarial attacks.  The results show that their method is as good as the SOTA, without relying on the same assumptions and approximations.  They also report that, from a latent perspective, there is no difference between the adversarial and OOD inputs; considering the input space was helpful in distinguishing between the two.

**Strengths:**

The question is important, the method is relatively straightforward, the main results are interesting, and there is good applicability to other work.  I like the idea of being able to filter inputs before feeding them into a larger model.

**Weaknesses:**

In general, I found the paper arguments difficult to follow. More scaffolding/introductory and connecting phrases would have been helpful.

**Questions:**

- In the discussion, the authors report "Even though adversarial examples from the discriminative model end up in the latent holes of the VAE, the active defense through HMC cannot return to the regions with high probability." Isn't this result in disagreement to the work by Kuzina et al., which suggests that you should be able to do this?  Beyond just differences in discriminative and latent models, is there any explanation for this?

---

### Comment · Area_Chair_RGzZ · 2024-11-22

Dear Authors and Reviewers,

The discussion phase has passed 10 days. If you want to discuss this with each other, please post your thoughts by adding official comments.

Thanks for your efforts and contributions to ICLR 2025.

Best regards,

Your Area Chair

---

### Note · Authors · 2024-11-25

**Comment:**

We are sincerely grateful to the reviewers for their thoughtful evaluation and constructive feedback on our submission. Their observations and suggestions have provided us with new perspectives and highlighted crucial areas for improvement in our work. After careful consideration, we have decided to withdraw our paper from the ICLR review process. While the current version of our manuscript supports the claims we initially presented based on a focused selection of datasets widely used in current literature, we have since conducted additional experiments and refined our analysis to address the points raised during the review process. We believe that incorporating this feedback has significantly strengthened our work, and we plan to submit an improved version to a different venue. Once again, we thank the reviewers for their valuable insights and the time they dedicated to reviewing our submission. Their thoughtful critique has helped us reframe and refine our research objectives.

**Withdrawal Confirmation:**

I have read and agree with the venue's withdrawal policy on behalf of myself and my co-authors.